# Coadjuvant Anti-VEGF A Therapy Improves Survival in Patients with Colorectal Cancer with Liver Metastasis: A Systematic Review

**Isabel Novo [1,†], Bárbara Campos [1,†], Filipa Pinto-Ribeiro [1,2] and Sandra F. Martins [1,2,3,*]**

1   School of Medicine, University of Minho, 4710-070 Braga, Portugal; a78995@alunos.uminho.pt (I.N.); a77840@alunos.uminho.pt (B.C.); filiparibeiro@med.uminho.pt (F.P.-R.)
2   Life and Health Sciences Research Institute (ICVS)/3B's - PT Government Associate Laboratory, 4710-070 Braga/Guimarães, Portugal
3   Surgery Department, Coloproctology Unit, Braga Hospital, 4710-243 Braga, Portugal
*   Correspondence: sandramartins@med.uminho.pt; Tel.: +351-253-604828; Fax: +351-253-604820
†   These authors contributed equally for the paper.

**Abstract:** Background: the presence of liver metastasis in colorectal cancer (CRC) remains one of the most significant prognostic factors. Objective: systematically review the results of studies evaluating the benefit of adding bevacizumab to a normal chemotherapy regime in the survival of patients with colorectal-cancer liver metastasis (CRLM). Search methods: Pubmed and Google Scholar databases were searched for eligible articles (from inception up to the 2 April 2019). Inclusion criteria: studies including patients with CRLM receiving anti-vascular endothelial growth factor (VEGF; bevacizumab) as treatment, overall survival as an outcome; regarding language restrictions, only articles in English were accepted. Main results: Eleven studies met the inclusion criteria. In 73% of these cases, chemotherapy with bevacizumab was an effective treatment modality for treating CRLM, and its administration significantly extended both overall survival (OS) and/or progression-free survival (PFS). Nevertheless, three articles showed no influence on survival rates of bevacizumab-associated chemotherapy. Author conclusions: It is necessary to standardize methodologies that aim to evaluate the impact of bevacizumab administration on the survival of patients with CRLM. Furthermore, follow-up time and the cause of a patient's death should be recorded, specified, and cleared in order to better calculate the survival rate and provide a comparison between the produced literature.

**Keywords:** anti-VEGF; colorectal cancer; hepatic metastasis; survival

## 1. Introduction

Colorectal cancer (CRC) represents a major public-health issue. There were more than 1.3 million new CRC cases in 2012, accounting for nearly 10% of the cancers [1]. It is now estimated that there are about 3.5 million people in the world with CRC, and that about 600,000 deaths occur each year (8% of all deaths from malignant neoplasms) [1]. It is the third most common cancer in men, and second in women, with an estimated CRC five-year survival rate of 65% in North America and 54% in Western Europe [2].

Liver metastasis from CRC remains one of the most significant prognostic factors for the treatment outcome. At the time of resection of the primary tumor, about 25% of patients with CRC already have liver metastases, and over the course of the disease, at least approximately 50% of patients develop liver metastasis.

CRC management has evolved into a multidisciplinary approach; when disease has spread to the surrounding lymph nodes, and even distant metastasis, a combination of chemotherapy, targeted

molecular-therapy agents, radiation, and surgical procedures for resectable metastasis have markedly improved overall survival (OS) [3]. Even so, surgical resection is the only potentially curative option in CRC liver-metastasis (CRLM) management [4].

Unfortunately, the major part of patients with CRLM are considered to be unresectable at presentation because of extrahepatic-disease involvement or insufficient remaining healthy liver tissue [5].

In order to solve this issue, there has been an increased use of chemotherapy before potentially curative surgery. Neoadjuvant chemotherapy has a number of potential advantages, including the possibility for previously unresectable tumors to become resectable; downsizing the tumor and increasing the chance of curative resection; determining tumor chemoresponsiveness to help select optimal adjuvant therapy, and identifying patients with a particularly aggressive disease in whom surgery would be inappropriate; in patients considered as resectable, neoadjuvant chemotherapy may treat undetected distant micrometastatic disease, thus reducing the risk of recurrence after resection.

Currently, treatment consists of a fluoropyrimidine doublet (FOLFOXCAPOX or FOLFIRI/CAPIRI) or fluoropyrimidine monotherapy (5-FU/folinic acid or capecitabine) in combination with a biological agent specific for vascular endothelial growth factor (VEGF) [6]. Current studies state that FOLFIRI and FOLFOX are identical in terms of efficacy, but vary in toxicity [6].

Monoclonal antibodies such as bevacizumab are part of the new generation of drugs in oncology, and have led to an improvement in disease prognosis, which has allowed the appearance of bases for research on the impact of biological therapy in CRLM treatment [7].

In the last few years, bevacizumab, a humanized immunoglobulin G1 monoclonal antibody targeting VEGF-A [8], was introduced in CRC treatment alongside chemotherapy [7]. It inhibits neovascularization, an essential mechanism for the growth and self-sustainability of tumors, and the vascular endothelial growth factor is one of the most important proangiogenic factors in both physiological and pathological conditions. When bevacizumab is coupled with VEGF, it deactivates its binding to the vascular endothelial growth factor receptor (VEGFR); therefore, it compromises the survival of cancer cells, inhibiting their growth [1]. However, further studies are needed to better understand the effective role of bevacizumab in survival, and its significance [1].

VEGF-A is the most studied member of the VEGF family, which also includes VEG-B, VEGF-C, VEGF-D, VEGF-E, and placental growth factor (PLGF).

The aim of this paper was to systematically review the results of studies evaluating the benefit of adding bevacizumab to a normal chemotherapy regime in the survival of patients with CRLM.

## 2. Results

Research summary is presented in Figure 1. We initially retrieved 258 potentially relevant reports, from which we excluded 238 articles for not meeting our language criterion, not being a primary study, not using anti-VEGF A, concerning other cancers types besides CRC, being animal studies, and not presenting survival analysis. The full-text reading excluded nine more articles due to lack of survival data.

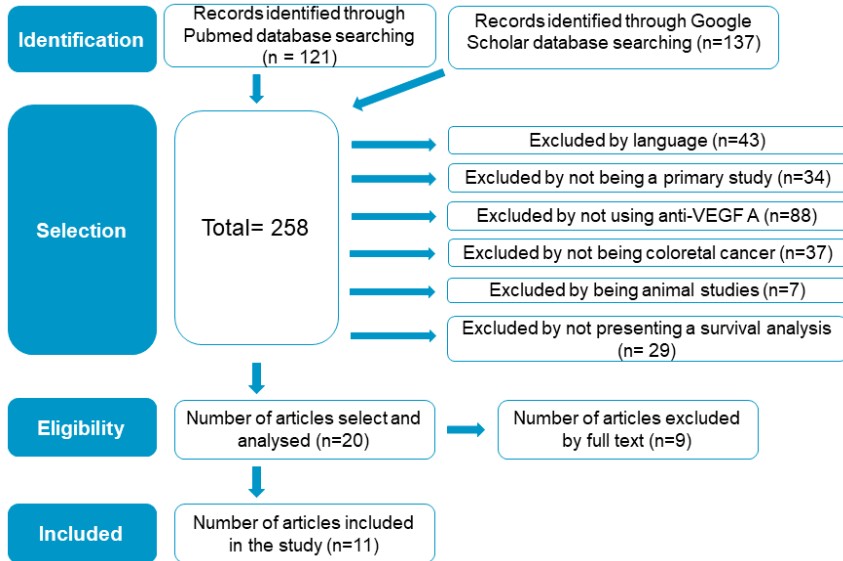

**Figure 1.** Flowchart following PRISMA (Preferred Reposting Item for Systematic Reviews and Meta-Analyses) statement.

## 2.1. Quality Assessment

The overview of study-quality analysis is presented in Figures 2–4. We divided this analysis into three parameters: high, moderate, and low quality.

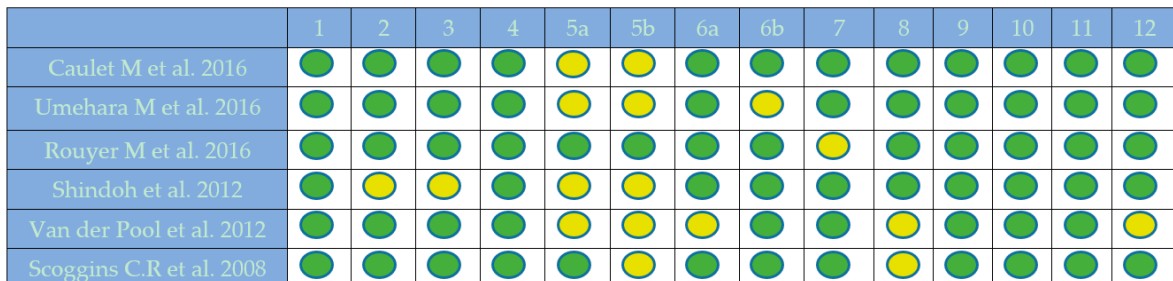

**Figure 2.** Quality evaluation of cohort studies based on Critical Appraisal Skills Programme (CASP) checklist. 1. Did the study address a clearly focused issue? 2. Was the cohort recruited in an acceptable way? 3. Was exposure accurately measured to minimize bias? 4. Was the outcome accurately measured to minimize bias? 5. (a) Have the authors identified all important confounding factors? 5. (b) Have they considered confounding factors in design and/or analysis? 6. (a) Was the follow-up of subjects complete enough? 6. (b) Was the follow-up of subjects long enough? 7. What are the results of this study? 8. How precise are the results? 9. Do you believe the results? 10. Can the results be applied to the local population? 11. Do the results of this study fit with other available evidence? 12. What are the implications of this study for practice? Note: green, yes; yellow, cannot tell.

Within the cohort studies, two of the six articles were classified as moderate-quality. The most failed criteria were related to the confounding factors in the study and how precise the results were.

In the clinical trials, all articles had high quality, although some parameters failed, such as the blindness of the study and the similarity between the groups at the start.

The case-control study had high quality, but some aspects were unclear with regard to the way the controls and cases were selected.

| | 1 | 2 | 3 | 4 | 5 | 6 | 7 | 8 | 9 | 10 | 11 |
|---|---|---|---|---|---|---|---|---|---|---|---|
| Chen H et al. 2016 | 🟢 | 🟢 | 🟢 | 🟡 | 🟢 | 🟢 | 🟢 | 🟢 | 🟡 | 🟢 | 🟢 |
| Beppu T et al. 2014 | 🟢 | 🟢 | 🟢 | 🟢 | 🟡 | 🟢 | 🟡 | 🟡 | 🟢 | 🟢 | 🟢 |
| Kemeny et al. 2010 | 🟢 | 🟢 | 🟢 | 🟡 | 🟡 | 🟢 | 🟢 | 🟢 | 🟢 | 🟢 | 🟢 |
| Gruenberger et al. 2008 | 🟢 | 🟡 | 🟢 | 🟡 | 🟢 | 🟢 | 🟢 | 🟢 | 🟢 | 🟢 | 🟢 |

**Figure 3.** Quality evaluation of clinical trials studies based on CASP checklist. 1. Did the trial address a clearly focused issue? 2. Was the assignment of patients to treatments randomized? 3. Were all of the patients who entered the trial properly accounted for at its conclusion? 4. Were patients, health workers, and study personnel 'blind' to treatment? 5. Were the groups similar at the start of the trial? 6. Aside from the experimental intervention, were the groups treated equally? 7. How large was the treatment effect? 8. How precise was the estimate of the treatment effect? 9. Can the results be applied to the local population or in your context? 10. Were all clinically important outcomes considered? 11. Are benefits worth the harm and costs? Note: green, yes; yellow, cannot tell.

| | 1 | 2 | 3 | 4 | 5 | 6a | 6b | 7 | 8 | 9 | 10 | 11 |
|---|---|---|---|---|---|---|---|---|---|---|---|---|
| Mahfud M et al. 2009 | 🟢 | 🟢 | 🟡 | 🟡 | 🟢 | 🟡 | 🟢 | 🟢 | 🟢 | 🟢 | 🟢 | 🟢 |

**Figure 4.** Quality evaluation of case-control study based on CASP checklist. 1. Did the study address a clearly focused issue? 2.Did the authors use an appropriate method to answer their question? 3. Were the cases recruited in an acceptable way? 4. Were the controls selected in an acceptable way? 5. Was exposure accurately measured to minimize bias? 6. (a) Aside from the experimental intervention, were the groups treated equally? 6. (b) Have the authors considered the potential confounding factors in the design and/or in their analysis? 7. How large was the treatment effect? 8. How precise was the estimate of the treatment effect? 9. Do you believe the results? 10. Can the results be applied to the local population? 11. Do the results of this study fit with other available evidence? Note: green, yes; yellow, cannot tell.

## 2.2. Population Description

This review included 11 studies that met our inclusion criteria with a total of population of 1345 patients, in which 785 are men and 560 are women. The age of the patients varied between 50 and 72 years. The studies used in this review were six cohorts, four clinical trials, and one case control. Most of them only analyzed CRLM, but when they included other types of metastasis, they were lung nodules, peritoneal nodules, or distant-lymph-node metastases.

## 2.3. Pharmacotherapy

The selected articles compared survival in patients with CRLM receiving treatment with bevacizumab associated with various chemotherapy regimens. As chemotherapy, 30% f the studies used mFOLFOX6, 30% FOLFOXIRI, and 30% oxaliplatin isolated with bevacizumab. In almost articles, the mean cycles of chemotherapy associated with bevacizumab were six. Eighty percent of the infusion of chemotherapy associated with bevacizumab was performed prior to resection of the tumor metastasis. In the different articles, different dosages of bevacizumab were used in the range of 5–7.5 mg/kg. In the study by Caulet et al. (2016) [9], the way to measure the amount of bevacizumab to be used was through its concentration in the serum, with one group receiving an amount of bevacizumab that allowed it to reach a concentration greater than 15.5 mg/L; in the other group, concentration in the serum was less than 15.5 mg/L.

Chemotherapy-associated bevacizumab appears to be an effective treatment modality for treating liver metastasis from colorectal cancer, and its administration significantly extends survival when compared with chemotherapy alone. It was generally agreed that bevacizumab inhibits tumor angiogenesis, decreases tumor-interstitial-fluid pressure that enables enhanced drug uptake by tumor cells, and protects against hepatic sinusoidal obstruction syndrome.

Regarding liver metastasis, one of the studies affirmed that complete remission was higher for those with exclusively hepatic metastasis than those with nonexclusively hepatic metastasis. So, bevacizumab therapy for CRLM allows besides better survival rates, better outcomes from resection of metastasis with complete remission, and it is a feasible, safe, and effective neoadjuvant regimen in patients with CRLM undergoing potentially curative liver resection [10].

However, one of the studies mentioned that the addition of bevacizumab to adjuvant hepatic arterial infusion (HAI) plus systemic therapy after liver resection did not seem to increase survival, but appeared to increase biliary toxicity [11].

## 2.4. Survival Outcome

Practically every study, more precisely 73% [4,5,9,10,12–15], concluded that bevacizumab therapy for the treatment of CRLM was effective and significant for both OS and/or progression-free survival (PFS), apart from three. These latest ones, groups receiving chemotherapy associated with bevacizumab, obtained worse survival rates than those who received the therapy without the drug being studied.

To evaluate OS, the different articles used several methods and measures. Ninety percent of the studies calculated the OS values, with an average of 32.5 months. In addition to this value, we were able to extract the survival values at three and five years from 55% of the articles, with a mean value of 80% and 56%, respectively. Only 40% of the authors collected PFS parameters, and the rest chose others: disease-free survival (DFS) in 30%, recurrence-free survival (RFS) in 10%, and the remaining only calculated either isolated OS or nothing, only concluding if the application of bevacizumab would or not have a positive effect on the patients.

In the trial by Caulet et al. (2016) [9], there were significant differences in median OS and median PFS between two groups with different doses of bevacizumab, these values being superior in the group whose dose was superior.

Chen H et al. (2016) [13] compared two groups: in Group 1, target vessel regional chemotherapy (TVRC) treatment was without bevacizumab; in Group 2, TVRC was associated with bevacizumab. Intrahepatic and extrahepatic time to progression was significantly different in the two groups, and much higher in the bevacizumab group. This indicated that bevacizumab could not only inhibit intrahepatic but also extrahepatic tumor vessels, thus postponing tumor progression. There was also a significant difference between the group that used bevacizumab and the one that did not in terms of survival.

Of the studies, 27% reported worse results in groups that used bevacizumab than those that did not, they being Van der Pool et al. (2012) [16], Kemeny et al. (2010) [11], and C. R. Scoggins et al (2009) [17]. In the first, the non-bevacizumab group (1) three-year DFS was 32%, and in the bevacizumab group (2) it was 23%, even though this result was not significant ($p = 0.35$). The second author confirmed that one-year RFS was 83% and 71%, and four-year RFS was 46% and 37% for non- bevacizumab (bev) and bev groups, respectively. The last one denoted that DFS of preoperative chemotherapy with the bevacizumab group [1] was 40 months, while the nonpreoperative-chemotherapy group [2] had median DFS of 56 months; again, these results were not significant ($p = 0.25$). In addition, OS was 56 months for Group 1 compared with 65 months for Group 2 ($p = 0.30$).

## 2.5. Included-Study Literature Search Results

The literature-search results, based on the applied keywords, and inclusion and exclusion criteria, are shown in Tables 1 and 2.

**Table 1.** Population description.

| Authors | Year | Study Type | Exclusion Criteria | Inclusion Criteria | No. of Patients | Age | No. Men | No. Women |
|---|---|---|---|---|---|---|---|---|
| Caulet et al. [9] | 2016 | Cohort | No info. | Eligible patients (18–80 years old) had histologically confirmed colorectal cancer (CRC) with at least one instance of hepatic metastasis detected by ultrasonography, life expectancy of more than two months, a World Health Organization (WHO) performance status of two or less, and were treated with first-line treatment by a bevacizumab-based chemotherapy. | 137 | 58–72 | 79 | 58 |
| Umehara et al. [4] | 2016 | Cohort | No info. | Patients were required to be between the ages of 18–80 years and to have histologically proven colorectal cancer with a World Health Organization performance status of two or less, potentially resectable liver metastases, and no detectable extrahepatic tumors. | 27 | 56–67 | 13 | 14 |
| Chen et al. [13] | 2016 | Clinical trial | Brain metastasis; ileus; inextricable obstructive jaundice; abdominal and pelvic effusion; and breastfeeding or pregnant women. | Patients diagnosed with colorectal cancer with liver metastasis; number of instances of liver metastasis of at least two or one, but difficult to resect by conventional surgery; after failure of first- or second-line or more systemic chemotherapy; expected lifetime of at least three months; no contraindication to treatment with chemotherapy; and age of at least 18 years old. | 63 | 40–80 | 41 | 22 |
| Rouyer et al. [10] | 2016 | Cohort | They were opposed to the collection of data regarding themselves or if they participated in a clinical trial, unless they were receiving a standard treatment (control arm) in an open-label Phase III study. | Proven diagnosis of CRC with nonresectable metastases, and treated with bevacizumab as first-line palliative therapy (nonresectability was documented in the medical files according to the multidisciplinary staff); when applicable, ≥6-month interval between adjuvant chemotherapy for primary tumor and initiation of bevacizumab; no chemotherapy for metastatic disease before the initiation of bevacizumab; and no prior history of treatment with bevacizumab, including as part of a clinical trial and during the period when bevacizumab was available under temporary use authorization (Autorisation Temporaire d'Utilisation). | 360 | 61–64 | 221 | 149 |

**Table 1.** *Cont.*

| Authors | Year | Study Type | Exclusion Criteria | Inclusion Criteria | No. of Patients | Age | No. Men | No. Women |
|---------|------|-----------|--------------------|--------------------|-----------------|-----|---------|-----------|
| Beppu et al. [14] | 2014 | Clinical trial | No info. | Patients with histologically proven colorectal cancer and at least one measurable lesion in the liver (with no nonhepatic distal metastasis/relapse) were eligible for this study if they met all of the following criteria: H2 or H3 CRC liver-metastasis (CRLM; either synchronous or metachronous); age ≥20 and ≤75 years; no prior chemotherapy except adjuvant chemotherapy if ended ≥6 months before study entry; no prior radiotherapy for advanced/recurrent colorectal cancer; Eastern Cooperative Oncology Group performance status (PS) 0 to 1; life expectancy estimated ≥ 3 months; adequate bone marrow and renal function. | 40 | 37–74 | 29 | 11 |
| Shindoh et al. [12] | 2012 | Cohort | No info. | No info. | 209 | 58 | 124 | 85 |
| Van der et al. [16] | 2012 | Cohort | Patients who had undergone portal-vein embolization (PVE) or portal-vein ligation (PVL), and patients who had been treated with other chemotherapeutics besides oxaliplatin-based chemotherapy. | Macroscopic radical resection of the liver metastases and the use of oxaliplatin-based CTx in neoadjuvant setting. | 104 | 41–79 | 62 | 42 |
| Kemeny et al. [11] | 2010 | Clinical trial | Extrahepatic disease, prior hepatic radiation, infection, history of stroke or transient ischemic attack, history of serious systemic illness, Karnofsky performance score <60, other malignancy (within five years before study entry), WBC count 3000/μL, absolute neutrophil count <1500/μL, platelet count ≤75,000/μL, and total bilirubin >2.0 mg/dL | Histologically confirmed colorectal adenocarcinoma with fully resected liver metastases. | 73 | 29 persons > or = to 60 years | 25 | 48 |

**Table 1.** *Cont.*

| Authors | Year | Study Type | Exclusion Criteria | Inclusion Criteria | No. of Patients | Age | No. Men | No. Women |
|---|---|---|---|---|---|---|---|---|
| Mahfud et al. [15] | 2009 | Case Control Study | No info. | Patients included were retrospectively assessed for eligibility using well-established databases in each respective center. | 90 | 54–65 | 50 | 40 |
| Scoggins et al. [17] | 2008 | Cohort | Patients treated with RFA only were not included. | Only patients who underwent hepatic resection or combined resection/radiofrequency ablation (RFA) for metastatic colorectal cancer between 1996 and 2006 were included in this analysis. | 186 | Preop.C group median age: 59 years versus 68.5 years in the NC group | 105 | 81 |
| Gruenberger et al. [5] | 2008 | Clinical trial | Prior chemotherapy for metastatic disease; prior history of bleeding diathesis or coagulopathy; clinical evidence of CNS metastases; history of thromboembolic or hemorrhagic events within 6 months before treatment; clinically significant cardiovascular disease. | Patients with histologically confirmed resectable CRC liver metastases who were at high risk of early recurrence defined as one or more risk factors according to Fong et al. These risk factors included: synchronous liver metastases; metastatic disease developed within one year after primary resection; lymph-node-positive primary tumors; more than one instance of liver metastasis; liver metastasis larger than 5 cm; and a positive carcinoembryonic antigen level. Patients were also required to have an Eastern Cooperative Oncology Group (ECOG) performance status of 0 to 1, adequate bone-marrow reserve, and adequate renal and hepatic function. | 56 | 61.5 | 32 | 24 |

**Table 2.** Study summary information.

| Authors | Year | Metastasis Tipe | Treatment | No. Chemocycles | Survival Outcome |
|---|---|---|---|---|---|
| Caulet M et al. [9] | 2016 | 75% liver metastasis only. | Bevacizumab + FOLFIRI (irinotecan, fluorouracil, leucovorin). | 4 | There were significant differences in overall survival (OS; P < 0.001) and in PFS (P = 0.001) between the two groups. OS was 17.3 months for patients with bevacizumab (Bev) lower than 15.5 mg/L, and 33.9 months for patients with Bev higher 15.5 mg/L. Median PFS (95% CI) was 8.7 months for patients with Bev lower than 15.5 mg/L, and 13.2 months for patients with Bev higher 15.5 mg/L. Increased drug exposure was shown to be associated with better clinical response and/or survival. |
| Umehara et al. [4] | 2016 | Liver metastasis only. | mFolfox6 (5-FU, leucovorin and oxaliplatin) + bevacizumab. | 6 | Three-year OS rate was 73.9%, and five-year OS rate was 62.5%. Disease-free survival (DFS) was significantly longer after mFOLFOX6 + Bev therapy compared to mFOLFOX6 therapy alone (p = 0.015). Optimal morphological response was associated with high five-year OS and DFS rates of 74% and 47%, respectively, and was observed in 47% of patients treated with bevacizumab and 12% of patients treated without bevacizumab. In the current study, we found no significant improvement in OS in patients treated with mFOLFOX6 + Bev compared to those treated with mFOLFOX6 alone (P = 0.23). Nevertheless, the observed increase in DFS was encouraging and significant (p = 0.015). |
| Chen et al. [13] | 2016 | Liver metastasis and extrahepatic metastasis. | Group 1: mFolfox6 + bevacizumab 100 mg. Group 2: mFolfox6 + bevacizumab 400 mg. | 6 | Two groups: in Group 1, target vessel regional chemotherapy (TVRC) treatment was without bevacizumab; in Group 2, TVRC was associated with bevacizumab. Time to progression of intrahepatic metastases (TTPIHM) was 3.53 and 6.23 (P = 0.018), and time to progression of extrahepatic metastases was 4.17 and 5.63 (P = 0.049) months in Groups 1 and 2, respectively. This indicated that bevacizumab could not only inhibit intrahepatic but also extrahepatic tumor vessels. and there was significant difference between groups in terms of survival. |
| Rouyer et al. [9] | 2016 | 52% liver metastasis only. | Irinotecan and bevacizumab. | Maximum of 8. | They observed that complete remission after surgery was high, in particular for patients with exclusively hepatic metastases, in line with induction chemotherapy associated with targeted therapy such as bevacizumab. These results support the idea that patients with exclusive hepatic metastases are better candidates to benefit from this type of curative treatment. |
| Beppu et al. [14] | 2014 | Liver metastasis only. | mFolfox6 (5-FU, leucovorin and oxaliplatin) + bevacizumab. | 5 | Stable disease was achieved in 55.0% of patients, and tumor-control rate was 85.0%. Median PFS of all patients (n = 40) was 9.7 months (95% CI = 6.2–11.8 months). Estimated one-, two- and three-year PFS was 35.0%, 22.5%, and 12.5%, respectively. Median survival time (MST) was 33.0 months (95%CI = 22.8 months—not reached). Estimated one-, two- and three-year OS was 87.5%, 62.3%, and 49.3%, respectively. |

Table 2. *Cont.*

| Authors | Year | Metastasis Tipe | Treatment | No. Chemocycles | Survival Outcome |
|---------|------|-----------------|-----------|-----------------|------------------|
| Shindoh et al. [12] | 2012 | Hepatic metastasis and extrahepatic lesions. | Oxaliplatin and bevacizumab. Irinotecan and bevacizumab. | 6 | This analysis determined that CT morphologic response to preoperative chemotherapy is a strong predictor of long-term outcomes after surgery in patients treated with or without bevacizumab. Optimal morphologic response was associated with high five-year OS and DFS rates of 74% and 47%, respectively. (95% CI: 1.46 to 4.49)On multivariate analysis, bevacizumab was strongly associated with optimal morphologic response (odds ratio, 6.71) |
| Van der et al. [16] | 2012 | Liver metastasis only. | Oxaliplatin and bevacizumab. | 4–6 | Estimated three-year disease-free survival in Groups 1 (non-bevacizumab) and 2 (bevacizumab) was 32% and 23%, respectively (P = 0.35). Bevacizumab added to oxaliplatin-based CTx may protect against moderate sinusoidal dilatation without significantly influencing morbidity. |
| Scoggins et al. [17] | 2010 | Liver metastasis only. | FOLFOXIRI (Irinotecan, oxaliplatin, 5-FU and leucovorin) plus systemic therapy with or without bevacizumab | 6 | Addition of Bev to adjuvant hepatic arterial infusion (HAI) plus systemic therapy after liver resection did not seem to increase RFS or survival but appeared to increase biliary toxicity. One-year RFS was 83% (95% CI, 66% to 92%) and 71% (95% CI, 52% to 83%), and four-year RFS was 46% and 37% (P=.4) for non-bev and bev arms. Four-year survival was 85% (95% CI, 60% to 95%) in the non-bev arm, and 81% (95% CI,56%to 93%) in the bev arm (log-rank test P = 0.5) |
| Mahfud et al. [15] | 2009 | Liver metastasis only. | FOLFOXIRI (Irinotecan, Oxaplatin, 5-FU and Leucovorin) plus systemic therapy with or without Bevacizumab | 8 | Overall morbidity rate was 56% (Bev) versus 40% (control); adjusted OR 1.74, 95% CI 0.71–4.28; p = 0.23. Mortality was 0 versus 2 in Bev and control groups, respectively. Causes of death entailed hepatic insufficiency with multiorgan failure. |
| Scoggins C.R et al. [17] | 2008 | Liver metastasis only. | FOLFOXIRI (irinotecan, oxaplatin, 5-FU, and leucovorin) plus systemic therapy with or without bevacizumab. | 8 | DFS and OS rates of preoperative-chemotherapy group were lower than in the nonpreoperative-chemotherapy group, although this was not statistically significant. DFS of the preoperative-chemotherapy group was 40 months, while the nonpreoperative-chemotherapy group had a DFS of 56 months. OS was 56 months for the preoperative-chemotherapy group compared with 65 months for the nonpreoperative-chemotherapy group. Five-year DFS and OS for the entire cohort were 48% and 53%, respectively. |
| Gruenberger et al. [5] | 2008 | Liver metastasis only. | Oxaplatin + bevacuzimab. | 6 | A total of 41 patients responded (objective response rate, 73.2%); five patients had a complete pathologic response (8.9%), and 36 had a partial response. Twelve additional patients (21.4%) had stable disease, and overall disease-control rate was 94.6%. This trial showed that perioperative chemotherapy followed by surgery and postoperative chemotherapy significantly improved three-year PFS in all eligible patients (36.2% v 28.1%; P 0.041). |

## 3. Discussion

This review included 11 studies with a total of population of 1345 patients, in which 785 are men and 560 are women. The age of the patients varied between 50 and 72 years. Most of them only analyzed CRLM, but some had extrahepatic metastasis such as lung nodules, peritoneal nodules, or distant-lymph-node metastasis. Regarding the quality of the studies, two of the eleven articles were classified as moderate-quality, and the other nine as high-quality.

The selected articles compared survival in patients with CRLM receiving treatment with bevacizumab associated with various chemotherapy regimens. More than 50% of patients with CRC develop liver metastases over the course of their pathology, which leads to the death of more than two-thirds of these patients [18,19]. Liver-resection surgery in patients with isolated metastatic liver disease remains the only option for a possible cure. However, even when resection is associated with modern systemic adjuvant regimens, it is only curative in 20% of patients [18–20]; 70% of patients experience disease recurrence, mainly in the liver [18].

Considering the mechanism of action of bevacizumab, almost all articles concluded that this drug improves prognostic survival by inhibiting tumor angiogenesis, a mechanism that plays an important role in malignant-tumor growth and persistence. Therefore, it compromises the survival of cancer cells, blocking their growth. It also decreases tumor interstitial fluid pressure that enables enhanced drug uptake by tumor cells and leads to an overgrowth of fibrosis at the expense of viable tumor cells.

In 73% of the included articles, chemotherapy with bevacizumab appeared to be an effective treatment modality for treating CRLM, and its administration significantly extended survival. In this review, we saw that the dose of bevacizumab could interfere with survival rate; a higher dose of this drug has better results in terms of efficacy and is associated to a longer survival. This suggests that increasing exposure to bevacizumab may increase its efficacy, and it is associated with a better clinical response of CRLM patients.

For example, a small Phase II study of bevacizumab combined with 5-FU and leucovorin suggested that a bevacizumab dose of 10 mg/kg was more effective than 5 mg/kg. Bevacizumab concentration higher than 15.5 mg/L was also associated with almost doubled OS when compared with patients whose concentration was below that value (33.9 vs. 15.5 months; $p = 0.006$) and PFS (13.2 vs. 8.7 months; $p = 0.0039$) [9]. These findings are in accordance with other analyses that designated that the rate of optimal morphologic response was moderately low (12%) in patients treated without bevacizumab when in comparison with patients treated with bevacizumab (47%); this means that it is toughly related with prime morphologic response (odds ratio, 6.71) [12].

On the other hand, not all articles showed a positive outcome in terms of survival. In 27% of the reviews, the groups receiving chemotherapy associated with bevacizumab did not obtain positive survival-outcome rates. In fact, those who received the therapy without the drug being studied had better prognosis. Despite this, these results were not significant, that is, although they did not improve the outcome, they did not worsen it either. Studies and results demonstrated by these articles were the first to be carried out, and, in the last decade, new innovative discoveries emerged. So, we concluded that the evolution of the investigation in this area shows the administration of bevacizumab has a positive impact on patient survival. In this group of articles, one of them was classified as moderate-quality.

Not all studies showed their follow-up time, which was important information to compare all of them, since it could have a huge effect on survival rate. Moreover, when mentioned, it would be similarly impossible to compare considering that different studies used different follow-up times. About the treatment itself, it was not uniform analysis in terms of when chemotherapy was administered (after or before surgery). In addition, not all studies performed resection of liver metastasis, which compromised the comparison of the studies, since this surgery is seen as the most effective treatment for CRLM, and potentially the only curative one.

When comparing the results of the studies, we noticed some difficulties because they used different measures in order to evaluate survival. Most studies did not clearly define if CRC metastasis was

synchronous or metachronous, isolated or with the involvement of another organ; they also did not clearly define the outcomes, namely, OS, PFS, and DFS. This may constitute a bias due to not being possible to compare those measures between them because they evaluate different parameters. One article did not measure survival, but just indicated if the addition of bevacizumab was beneficial or not in terms of overall disease control. In addition, not all studies defined the specific cause of death of the patient when they evaluated survival, so it cannot be confirmed that patients died only due to the CRLM, or if death was from other causes. Bevacizumab has also never been studied in isolation, but always associated with other drugs.

It is necessary to standardize the methodologies of studies that aim to evaluate the impact of the administration of bevacizumab on the survival of patients with CRLM. Furthermore, it is necessary to define the parameters to be evaluated in each study, defining, for example, the follow-up time and the cause of the patient's death in order to better calculate the survival rate, and provide a comparison between the produced literature.

Bevacizumab plays an important role in the treatment of CRLM, and it can improve patient survival, although there were three studies showing no positive outcomes. We conjectured that it is necessary to standardize and universalize some of the most important and essential parameters for the evaluation of survival as a main outcome in studies of this kind, such as overall survival, follow-up time and if this was performed in the most correct way, if patients in had comorbidities that would influence their prognosis, and the specific cause of death in order to produce a more solid literature. This area requires more studies, and it would be advantageous and enriching to carry out future meta-analysis on this topic to evaluate the concordance and relationship between various studies.

*Recommendations*

Chemotherapy associated with bevacizumab is increasingly used in association with surgery, not only in the neoadjuvant treatment of patients with liver metastases that would initially be unresectable in order to make them resectable, but also in patients with resectable liver metastases with the objective of reducing the risk of recurrence.

As a guideline, in patients with resectable CRLM, perioperative chemotherapy (before and after resection surgery) has become the standard treatment in many studies. In patients with unresectable CRLM at diagnosis, the combination of neoadjuvant chemotherapy with bevacizumab and surgery is a good way to increase the survival of these patients and improve their prognosis. However, it is essential that these patients are carefully monitored, and surgery is performed as soon as metastases become resectable.

On the other hand, there are several questions that remain unanswered and that still need reliable studies, such as how long before surgery to perform neoadjuvant chemotherapy, what the ideal number of cycles are, and what the interval between surgery and the new treatment of chemotherapy is. We also know that a higher dose of bevacizumab is associated with better a clinical outcome, but we do not objectively know what the most correct dose is, considering that it was not uniform in all studies.

Nevertheless, there is a parameter that is essential, particularly at a time when more combinations of chemotherapeutic agents, and more developed surgical procedures and complementary diagnostic tests are appearing: the therapeutic scheme and monitoring of each patient must be carefully decided and studied by competent and dedicated multidisciplinary teams.

## 4. Materials and Methods

*4.1. Criteria for Considering Studies for this Review*

### 4.1.1. Types of Studies

Study that analyzed the effects on survival of adding bevacizumab to chemotherapy in patients with CRLM.

### 4.1.2. Types of Participants

Studies including participants with colorectal cancer with at least liver metastasis.

### 4.1.3. Types of Interventions

Trials investigating treatment with bevacizumab at any dosage and in any formulation.

### *4.2. Search Methods for Identification of Studies*

### 4.2.1. Data Sources and Searches

A systematic review of the literature was made following a predefined protocol, with the objective of identifying studies appraising and comparing the survival of patients with colorectal cancer and liver metastasis who were treated with chemotherapy with bevacizumab.

The Pubmed and Google Scholar databases were searched for eligible articles (from inception up to 2 April 2019). The following keywords were used in the search: "anti-VEGF", "colorectal cancer", "hepatic metastasis", and "survival". If there were repeated publications, we selected the most recent articles and those that presented more data. Ultimately, we used the "snowball" method aiming to identify additional relevant papers by tracking the references and citations of found articles.

### 4.2.2. Selection of Studies, and Data Extraction and Management

Two of the review authors (I.N. and B.C.) independently evaluated all yielded titles and abstracts for eligibility. We resolved disagreements by consensus or by involvement of a third review author (S.F.M.). In the case of a few articles describing the same trial, we selected the conclusive article as the main report, and analyzed the remaining articles for complementary information on clinical outcomes, descriptions of study participants, or design characteristics.

We extracted the type of control used, author names, year of publication, sample size, primary endpoint, results on OS and PFS, and, when these data were not available, we extracted disease-free survival (DFS) or anything else that gave us information on survival, such as dosage and number of bevacizumab cycles, and certain participant characteristics such as age and gender. We also extracted important conclusions from the publication status.

### 4.2.3. Study-Eligibility Criteria

Studies investigating the associations between anti-VEGF A (bevacizumab) and OS in patients with hepatic metastatic colorectal cancer were initially reviewed. Two of the review authors (I.N. and B.C.), working separately and in parallel, checked the abstracts, and then fully accessed and reviewed only articles that met the established inclusion and exclusion criteria.

Inclusion criteria: studies including patients with CRLM, patients receiving anti-VEGF (bevacizumab) as treatment, overall survival as an outcome; regarding language restrictions, only articles in English were accepted.

Exclusion criteria: types of paper that were not primary studies, studies that did not have patients assigned to anti VEGF-A (bevacizumab), not having colorectal cancer with liver metastasis; papers that did not measure survival as an outcome, studies not performed on humans, and papers that were not in the English language.

### 4.2.4. Quality Assessment of Included Studies

Two of the review authors (I.N. and B.C.) independently assessed the adequacy of randomization, blinding, and analyses, verifying methodological validity for every study that met the inclusion criteria. Any disagreements were resolved through consensus-based discussion or with a third review author (S.F.M.). Quality assessment of the included studies was based on recommendations given by the Critical Appraisal Skills Program (CASP) checklists for cohort [21], randomized controlled Trials [22],

and case-control [23] studies. On the basis of information given by the journal publications, we judged each domain in each study as high-, moderate- and low-quality.

### 4.2.5. Data Synthesis

The primary endpoints were PFS and OS. OS was characterized as the length of time since randomization to the occurrence of death. PFS illustrated the length of time since randomization to the first occurrence of progression, relapse, or death from any cause. When these were not available, we extracted the DFS, which was measured from the date of hepatic resection until the date of radiographic detection of recurrence or last follow-up. Benefits in OS and PFS from adding bevacizumab to standard chemotherapy regimens had their values assessed as a percentage, their respective 95% confidence intervals (CI), and the published Kaplan–Meier curves to collect any data that would help us understand the influence of this therapy on survival.

## 5. Conclusions

Our data suggested that bevacizumab plays an important role in the treatment of CRLM, as recommended by international guidelines. To obtain fewer conflicting data, it is necessary that these studies clearly define the parameters to be evaluated and to standardize the methodologies of the studies.

**Author Contributions:** conceptualization, I.N., B.C., F.P.-R., and S.F.M.; methodology, I.N., B.C., F.P.-R., and S.F..M; validation, F.P.-R. and S.F.M.; formal analysis, I.N. and B.C.; investigation, I.N. and B.C.; writing—original-draft preparation, I.N. and B.C.; writing—review and editing, I.N., B.C., F.P.-R., and S.F.M.; supervision, S.F.M.; project administration, S.F.M. All authors have read and agreed to the published version of the manuscript.

**Funding:** This research received no external funding

**Conflicts of Interest:** The authors declare no conflict of interest.

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
