# Peer review of "Coadjuvant Anti-VEGF A Therapy Improves Survival in Patients with Colorectal Cancer with Liver Metastasis: A Systematic Review"

_gastrointestdisord, doi:10.3390/gidisord2020007_

Round 1

Reviewer 1 Report

Thank you very much for your interesting work. However, there are some points which should be cleared prior to publication:

The authors should state more clearly what the focus and goal of their work is, as Bevacizumab is an established therapy for mCRC. Its use is recommended by international guidelines. This should be reflected more in the manuscript.

The authors cite a meta-analysis of Bevacizumab therapy in mCRC (doi:10.1371/journal.pone.0161912). It analyzes randomized phase II + III studies which are not included in the study by the authors. To answer the author's question it is probably necessary to extract the data from the patients with liver metastases from these studies which have a high level of evidence.

Concerning the presentation of the studies in the tables 1 & 2 the use of full sentences renders the information difficult to read.

The manuscript is well written and structured, but the authors's conclusion that "Bevacizumab plays an important role in the treatment of CRLM" is foreseeable given the recommendation of its use in international guidelines. From the data presented it seems difficult to draw any conclusions about the impact on survival as no structured analysis is shown.

Author Response

Comment#1: The authors should state more clearly what the focus and goal of their work is, as Bevacizumab is an established therapy for mCRC. Its use is recommended by international guidelines. This should be reflected more in the manuscript.

Response: We thank the reviewer for the comment, we have reformulated this issue in the following manner: “The aim of this paper was to systematically review the results of studies evaluating the benefit of adding bevacizumab to the normal chemotherapy regime in the survival of patients with CRLM”.

Comment#2: The authors cite a meta-analysis of Bevacizumab therapy in mCRC (doi:10.1371/journal.pone.0161912). It analyzes randomized phase II + III studies which are not included in the study by the authors. To answer the author's question it is probably necessary to extract the data from the patients with liver metastases from these studies which have a high level of evidence.

Response: We thank the reviewer for the comment however we would like to recall the reviewer’s attention to the fact that we proposed to perform a systematic review of the literature, thus due to the nature of our work we can only include primary studies in our results section and not reviews. In addition, in the above mentioned review the focus was not restricted to patients presenting liver metastases but included all patients with metastases

Comment#3: Concerning the presentation of the studies in the tables 1 & 2 the use of full sentences renders the information difficult to read.

Response: We thank the reviewer for the comment however we think that using uncomplete sentences would perform a lack of information about our work. Furthermore, we have formatted the tables in a way to be easier to read them.

Comment#4: The manuscript is well written and structured, but the authors's conclusion that "Bevacizumab plays an important role in the treatment of CRLM" is foreseeable given the recommendation of its use in international guidelines. From the data presented it seems difficult to draw any conclusions about the impact on survival as no structured analysis is shown.

Response: We thank the reviewer for the comment, we had reviewed our conclusion. “Our data suggests Bevacizumab plays an important role in the treatment of CRLM, as recommended by international guidelines.  To obtain less conflicting data, it is necessary that these studies clearly define the parameters to be evaluated and to standardize the methodologies of the studies.”

Reviewer 2 Report

The authors have conducted a systematic review examining the efficacy of anti-VEGF therapy added to chemotherapy in patients with colorectal liver metastasis. This is an important study as a large portion of the population suffers from CRC and liver mets are a poor prognostic indicator. The authors have done a thorough evaluation of published clinical studies and summarized their findings well. A few changes would improve the quality of the paper.

Introduction
Line 40 is unclear needs rephrasing.

There are multiple VEGF rectors the authors do not indicate this.

Line 62 Consider revising

VEGF-A is mentioned in the results but not in the intro

A discussion on neoadjuvant or adjuvant setting and operable and inoperable cases is needed

Results
Line 65 typos

Line 98 – extended survival over what? Chemo alone? This needs to be clearly stated

Line 99 – “this drug” should read bevacizumab

Line 115 – is not clearly stated, consider revising

Discussion

A graphic or list of recommendations would be beneficial here

More information on the effects of Beva dosing is needed in the results to inform this part of the discussion

Overall

There are many English grammar issues and typos throughout. Please have this edited before publication

Author Response

Comment#1: Line 40 is unclear needs rephrasing.

Response: We thank the reviewer for the comment we have rephrased the sentence. “At the time of resection of the primary tumor, about 25% of patients with CRC already have liver metastases and it is known that over the course of the disease at least approximately 50% of patients will develop liver metastasis.”

Comment#2: There are multiple VEGF rectors the authors do not indicate this.

Response: We thank the reviewer for the comment, we have corrected this part in the following manner: “VEGF-A is the most studied member of the VEGF family. The VEGF family also include the VEG-B, VEGF-C, VEGF-D, VEGF-E and placental growth factor (PLGF).”

Comment#3: Line 62 Consider revising

Response: We thank the reviewer for the comment we have reviewed the sentence: “The aim of this paper was to systematically review the results of studies evaluating the benefit of adding bevacizumab to the normal chemotherapy regime in the survival of patients with CRLM.”

Comment#4: VEGF-A is mentioned in the results but not in the intro

Response: We thank the reviewer for the comment we have included this information on the introduction as suggested “VEGF-A is the most studied member of the VEGF family”

Comment#5: A discussion on neoadjuvant or adjuvant setting and operable and inoperable cases is needed.

Response: We thank the reviewer for the comment, we have included this information on the introduction as suggested. “Unfortunately, the major part of the patients with CRLM are considered to be unresectable at the presentation because of the extrahepatic disease involvement or the insufficient remaining heathy liver tissue.

In order to solve this issue, there has been an increased use of chemotherapy before the potentially curative surgery. Neoadjuvant chemotherapy has number of potential advantages including may allow for previously unresectable tumors to become resectable; may down-size the tumor and increase the probability of curative resection; may be useful top determine chemoresponsiveness of the tumor to help select the optimal adjuvant therapy, as well as identify patiens with particularly aggressive disease in whom surgery would be inappropriate; and in patients considered as resectable, neoadjuvant chemotherapy may treat undetected distant micrometastatic disease, thus, reducing the risk of recurrence after resection.”

Comment#6: Results - Line 65 typos

Response: We thank the reviewer for the comment we haver reviewed the sentence. “A summary of the research is presented in figure 1”.

Comment#7: Line 98 – extended survival over what? Chemo alone? This needs to be clearly stated

Response: We thank the reviewer for the comment we have clarified the sentence. “Chemotherapy associated bevacizumab appears to be an effective treatment modality for treating liver metastasis from colorectal cancer, and its administration significantly extends survival when compared with chemotherapy alone”.

Comment#8: Line 99 – “this drug” should read bevacizumab

Response: We thank the reviewer for the comment we have corrected the sentence. “It was generally agreed that bevacizumab inhibits tumour angiogenesis, decreases tumour interstitial fluid pressure that enables enhanced drug uptake by tumour cells and protects against hepatic sinusoidal obstruction syndrome”

Comment#9: Line 115 – is not clearly stated, consider revising

Response: We thank the reviewer for the comment we have reviewed the sentence. “These latest ones, the groups receiving chemotherapy associated with bevacizumab obtained worse survival rates than those who received the therapy without the drug being studied.”

Comment#10: Discussion - A graphic or list of recommendations would be beneficial here

Response:  We thank the reviewer for his comment, it really makes sense to put that aspect into the work and it had not occurred to us before. We added this to our discussion: Chemotherapy associated with Bevacizumab is being used more and more in association with surgery, not only in the neoadjuvant treatment of patients with liver metastases that would initially be unresectable in order to make them resectable, but also in patients with resectable liver metastases, with the objective of reducing the risk of recurrence.

As a guideline, in patients with resectable CRLM, perioperative chemotherapy (before and after resection surgery) has become the standard treatment in many of our studies. In patients with unresectable CRLM at diagnosis, the combination of neoadjuvant chemotherapy with Bevacizumab and surgery is a good way to increase the survival of these patients and also their prognosis. However, it is essential that these patients are carefully monitored and the surgery is performed as soon as the metastases become resectable.

On the other hand, there are several questions that remain unanswered and that still need reliable studies, such as how long to perform neoadjuvant chemotherapy before surgery, what is the ideal number of cycles and what is the interval between surgery and the new treatment of chemotherapy. We also know that a higher dose of Bevacizumab is associated with better clinical outcome, however we do not know objectively what the most correct dose will be, considering that it was not uniform in all studies.

Nevertheless, there is a parameter that is essential, particularly at a time when more and more combinations of chemotherapeutic agents, more developed surgical procedures and complementary diagnostic tests are appearing, the therapeutic scheme and the monitoring of each patient must be carefully decided and studied by competent and dedicated multidisciplinary teams.

Comment#11: More information on the effects of Beva dosing is needed in the results to inform this part of the discussion

Response: We thank the reviewer for the comment we have added more information about the dosage of Bevacizumab as suggested. “In the different articles, different dosages of bevacizumab were used, ranging from 5-7.5 mg / kg. In the study by Caulet et al. (2016) [9] the way to measure the amount of Bevacizumab to be used was through its concentration in the serum, with one group receiving an amount of bevacizumab that allowed it to reach a concentration greater than 15.5 mg / L and in the other group the concentration in the serum was less than 15.5 mg/L.”

Comment#12: Overall - There are many English grammar issues and typos throughout. Please have this edited before publication

Response: We thank the reviewer for the comment, and we have already corrected and edited the text.

Reviewer 3 Report

The systematic review from Novo et al. is about the improvement of survival in colorectal cancer with liver metastasis of patients treated with Bevacizumab, an anti-VEGF A, in combination with different chemotherapies. The present review includes the analyses of either cohort studies, case control studies and randomized clinical trials, each well described. Results and conclusions are clearly presented. Inclusions and exclusions criteria for the identified articles included in the present study are well described and seem to be relevant. Despite the manuscript is suitable for publication in Gastrointestinal Disorders I thereby recommend to change the title to avoid confusion for the readers, as Bevacizumab was not assessed as a single treatment but was evaluated in combinations with different chemotherapies. Especially considering the fact that resection of liver metastasis was also performed in some of the included studies. The sentence L23-25 of the abstract should be replace by something like: “However, three articles documented no influence on survival rates of Bevacizumab combined chemotherapy” as the current wording makes totally no sense compared to the previous sentence.

Author Response

Comment#1: Despite the manuscript is suitable for publication in Gastrointestinal Disorders I thereby recommend to change the title to avoid confusion for the readers, as Bevacizumab was not assessed as a single treatment but was evaluated in combinations with different chemotherapies.

Response: We thank the reviewer for the comment we have replaced the title for “Coadjuvant Anti-VEGF A therapy improves survival in patients with colorectal cancer with liver metastasis: A systematic review.”

Comment#2: Especially considering the fact that resection of liver metastasis was also performed in some of the included studies. The sentence L23-25 of the abstract should be replace by something like: “However, three articles documented no influence on survival rates of Bevacizumab combined chemotherapy” as the current wording makes totally no sense compared to the previous sentence.

Response: We thank the reviewer for the comment we have replaced the sentence as suggested. “Nevertheless, three articles showed no influence on survival rates of Bevacizumab associated chemotherapy”.

Reviewer 4 Report

This current article reviews the benefit of adding Anti-VEGF (Bevacizumab) to conventional chemotherapy in the treatment of colorectal cancer liver metastasis (CRLM). While colon cancer is one of the most common cancer worldwide, it is important to established an protocol that covers neoadjuvant chemotherapy, adjuvant and surgery therapy. In all malignancies, early detection and a solid treatment, is a key factor for increasing the survival rate. Liver metastasis from CRC remains one of the most significant prognostic factors for treatment outcome. We all known that in the moment of the surgery 25% of CRC patients have hepatic metastasis and approximately 50% of CRC patients will develop liver metastasis during their disease. It is important to have a multidisciplinary approach, when the disease has spread into the body, and this approach includes a combination of chemotherapy, molecular agents, radiation and surgical procedures for resectable metastasis. With all this methods of treatment we need to improve overall survival (OS).  This paper performs a literature review of studies, which analysethe survival benefit of adding Bevacizumab to chemotherapy in patients with CLRM. Bevacizumab is a recombinant, humanized antibody against vascular endothelial growth factor (VEGF) that is used to inhibit VEGF function in vascular endothelial cells and therapy inhibits tumor angiogenesis, upon which solid tumors depend for growth and metastasis. At present the addition of Bevacizumab to fluoropyrimidine-based chemotherapy, in both the first and second line treatment of metastatic colorectal cancer, significantly increase median progression-free survival or time to disease progression in most randomized controlled trials. Bevacizumab had acceptable tolerability, with the majority of adverse events being generally mild and clinically manageable. The standard therapy for CLRM include mFOLFOX, FOLFOXIRI and Oxaplatin, and in most articles, the mean cycles of chemotherapy associated with Bevacizumab were six. Regarding the liver metastasis, there are evidence that in patients which had Bevacizumab as a neoadjuvant monotherapy or combinations the complete remission was higher and had better outcomes for resection of metastasis.

Most of the articles concluded that Bevacizumab therapy for the treatment for CRLM was effective and significant for both OS and/or PFS, and most of them are high quality studies, but even with this modern therapy, surgery remains the only radical therapy for CRLM but with a high mortality rate, and when resection is combined with modern adjuvant systemic regimens, it is curative in only 20%. Bevacizumab has an important role in the treatment of CRLM and it can improve the patients survival but it is also important to mention that this area requires more studies and it would be advantageous to carry out a future meta-analysis on this topic.

The present article is written in a clear and concise manner, analyzing data from literature with precision and clarity, proving to be valuable asset in it’s field.

Author Response

We thank the reviewer for the comment.

No changes were requested.

Round 2

Reviewer 1 Report

Thank you for the revised version of the manuscript, which has considerably improved.

I am well aware that only primary studies are suitable for inclusion in a systematic review. This is why I recommended extracting data from the patients with liver metastases from the primary studies itself and not including the meta-analysis. However, I acknowledge the difficulties in acquiring the data and think, the manuscript is suitable for publication.